# The Third Dose of BNT162b2 COVID-19 Vaccine Does Not “Boost” Disease Flares and Adverse Events in Patients with Rheumatoid Arthritis

**DOI:** 10.3390/biomedicines11030687

**Published:** 2023-02-23

**Authors:** Andrea Picchianti Diamanti, Assunta Navarra, Gilda Cuzzi, Alessandra Aiello, Simonetta Salemi, Roberta Di Rosa, Chiara De Lorenzo, Daniele Vio, Giandomenico Sebastiani, Mario Ferraioli, Maurizio Benucci, Francesca Li Gobbi, Fabrizio Cantini, Vittoria Polidori, Maurizio Simmaco, Esmeralda Cialdi, Palma Scolieri, Vincenzo Bruzzese, Emanuele Nicastri, Raffaele D’Amelio, Bruno Laganà, Delia Goletti

**Affiliations:** 1Department of Clinical and Molecular Medicine, Sapienza University of Rome, S. Andrea University Hospital, 00189 Rome, Italy; 2Translational Research Unit, National Institute for Infectious Diseases Lazzaro Spallanzani-IRCCS, 00161 Rome, Italy; 3Department of Rheumatology, San Camillo Hospital, 00152 Rome, Italy; 4Rheumatology, Allergology and Clinical Immunology, Department of ‘Medicina dei Sistemi’, University of Rome ‘Tor Vergata’, 00133 Rome, Italy; 5Rheumatology Unit, S. Giovanni di Dio Hospital, Azienda USL-Toscana Centro, 50122 Florence, Italy; 6Rheumatology Department, Hospital of Prato, 59100 Prato, Italy; 7UOC Laboratorio Analisi e Biochimica Clinica, Sant’Andrea University Hospital, 00189 Rome, Italy; 8UOC di Medicina e Rete Reumatologica, Ospedale Nuovo Regina Margherita, 00153 Rome, Italy; 9Clinical Division of Infectious Diseases, National Institute for Infectious Diseases Lazzaro Spallanzani-IRCCS, 00161 Rome, Italy

**Keywords:** COVID-19, SARS-CoV-2 vaccine, immunogenicity, adverse events, disease flare, autoimmune rheumatic diseases

## Abstract

Data on the risk of adverse events (AEs) and disease flares in autoimmune rheumatic diseases (ARDs) after the third dose of COVID-19 vaccine are scarce. The aim of this multicenter, prospective study is to analyze the clinical and immunological safety of BNT162b2 vaccine in a cohort of rheumatoid arthritis (RA) patients followed-up from the first vaccine cycle to the third dose. The vaccine showed an overall good safety profile with no patient reporting serious AEs, and a low percentage of total AEs at both doses (40/78 (51.3%) and 13/47 (27.7%) patients after the second and third dose, respectively (*p* < 0.002). Flares were observed in 10.3% of patients after the end of the vaccination cycle and 12.8% after the third dose. Being vaccinated for influenza was inversely associated with the onset of AEs after the second dose, at both univariable (*p* = 0.013) and multivariable analysis (*p* = 0.027). This result could allow identification of a predictive factor of vaccine tolerance, if confirmed in larger patient populations. A higher disease activity at baseline was not associated with a higher incidence of AEs or disease flares. Effectiveness was excellent after the second dose, with only 1/78 (1.3%) mild breakthrough infection (BI) and worsened after the third dose, with 9/47 (19.2%) BI (*p* < 0.002), as a probable expression of the higher capacity of the Omicron variants to escape vaccine recognition.

## 1. Introduction

At the beginning of 2021, a vaccine campaign against COronaVIrus Disease 2019 (COVID-19) started worldwide, and patients affected by autoimmune rheumatic diseases (ARDs) were included among the priority target groups [1,2,3,4].

Indeed, patients with ARDs are known to have a higher risk of serious infections and a slightly higher risk of complications from COVID-19 than the general population [4,5,6,7].

Currently, several authors evaluated the immunogenicity of COVID-19 vaccine (mainly BNT162b2 vaccine) in ARD patients through the analysis of humoral and cellular immunity. Overall, these studies showed that the great majority of ARD patients develop anti-receptor-binding domain (RBD) antibodies, with a lower proportion presenting also a specific T-cell response [8,9,10]. The level of both humoral and cellular specific response is significantly reduced compared to healthy controls and the main factor affecting immunogenicity is the use of some immunosuppressive agents such as the anti-Cluster of Differentiation (CD)20, Cytotoxic T-Lymphocyte Antigen (CTLA)-4Ig and mycophenolate [8,9,10]. Furthermore, it has been shown that vaccine immunogenicity, in particular the level of specific antibodies, decreases over time [11], but it is significantly enhanced by a third vaccine dose (first booster) [12,13].

Inflammatory arthritis, and, in particular, Rheumatoid Arthritis (RA), is characterized by spontaneous and unpredictable episodes of immunological and clinical worsening, defined flare ups. They may frequently be induced by infections, which have even been implicated in the etiology of the disease and may trigger a strong inflammatory reaction able to reactivate the disease [14]. Thus, the prevention of infectious diseases is crucial for maintaining the state of drug-induced remission. However, the concern that even vaccines may mimic the inflammatory activity of microorganisms, thus being able to either inducing or reactivating ARDs [15,16], has traditionally hampered a large application of prophylactic vaccines in these patients [17]. The concern of disease reactivation has even increased with the innovative COVID-19 vaccines, due to a lack of ARD patients in the explored population during the pivotal trials and of long-term vaccine experience [18]. In these first two years of pandemic, different studies of vaccine safety have been developed in patients with ARDs, with RA being always well represented [19,20,21,22,23,24,25,26,27,28,29,30,31,32,33,34,35,36,37,38,39,40]. In fact, over 10.000 RA patients have been investigated for safety among the ARD population, but only four studies, including fewer than 1.000 RA patients, are exclusively focused on this disease. Moreover, most studies are affected by several limitations, having frequently a retrospective design, the occurrence of disease flares having been inferred through survey, or having used different study protocols, including different anti-COVID-19 vaccines or patients with different baseline disease activity, thus making them scarcely comparable. 

We carried out, therefore, a multicenter, prospective study to analyze the clinical and immunological safety of BNT162b2 vaccine, in a cohort of RA patients, in remission and with low-moderate disease activity, followed-up from the first vaccine cycle to the first booster dose.

## 2. Patients and Methods

### 2.1. Ethics

The protocol was approved by the Ethical Committee of INMI-Lazzaro Spallanzani-IRCCS (approval numbers 297/2021, 247/2021 and 318/2021). All participants signed a written informed consent.

### 2.2. Study Design

This is a prospective, real-life, multicenter study investigating the safety of mRNA vaccine (BNT162b2, Pfizer–BioNTech) in a cohort of RA patients during first vaccine cycle and third dose administration (first booster).

### 2.3. Study Population and Location

Patients with a diagnosis of RA according to the 2010 criteria of the European League Against Rheumatism/American College of Rheumatology (EULAR/ACR) [41], under immunosuppressive treatment, were enrolled at S. Andrea University Hospital in Rome, AO San Camillo Forlanini (Rome, Italy), Nuovo Regina Margherita Hospital (Rome, Italy), Rheumatology Department of Prato Hospital (Prato, Italy) and Rheumatology Unit of S. Giovanni di Dio Hospital (Florence, Italy).

### 2.4. Sample Size Estimation

We conducted post hoc power calculations at a two-sided 5% significance level to evaluate the ability of the study to detect the difference observed between two groups. Moreover, to address this issue, in logistic regression odds ratios with their 95% confidence interval (95% CI) were also reported as estimate of the magnitude of effects that are consistent with the observed data.

### 2.5. Components of Data Collection and Operational Definitions

At T0 (time of the first dose of vaccination), several demographic and clinical data including body mass index (BMI), smoking habits, comorbidities, flu vaccination, ongoing immunosuppressive therapy, disease duration and disease activity, were collected (Table 1). In particular, disease activity was assessed by using the Disease Activity Score on 28 joints (DAS28CRP).

At T1 (2–4 weeks from the second vaccine dose) and T2 (4–6 weeks from the booster dose, which was administered in November and December 2021), the following parameters were assessed:− DAS28CRP to infer for possible disease flare-ups. Flare-up was defined as ΔDAS28 ≥ 1.2 or ≥0.6, if final DAS28 > 3.2 [42]. The duration of flare, temporary suspension of immunosuppressive therapy during vaccination, and any action taken to manage the disease flare was also registered.− AEs collected as early local and systemic (reactogenicity), if occurred within 7 days from vaccination: pain, redness or swelling at the site of injection; generalized muscle or joint pain, headache, fever, chills, fatigue or vomiting. AEs of special interest, reported by specifying the organ/system involved. It has to be underlined that most systemic AEs, excepting for anaphylactic reactions, are closely evocative of flares; thus, they may hardly be discriminated from them. In this study, however, it was preliminarily established that systemic AEs were considered flares only if they were able to increase the DAS28CRP as described above, whereas, in case of mild symptomatology, they were categorized as AEs. AEs were also classified as non-serious or serious (SAEs, requiring hospitalization, being life-threatening, persistent or leading to significant disability or resulting in death);− Antinuclear antibodies (ANA) and rheumatoid factor (RF) in a subgroup of patients by using indirect immunofluorescence on hep-2 cells, and nephelometry, respectively;− Occurrence of breakthrough infections (BI) 

The patients interrupted methotrexate and JAK-inhibitors intake for one week before and after, and CTLA-Ig one week before, each vaccine dose as suggested by ACR recommendations [7].

### 2.6. Statistical Analysis

Numerical variables were expressed as median and interquartile range (IQR) and compared among groups by means of the Mann–Whitney test. Categorical variables were expressed by numbers and percentages and compared among groups by Yates corrected, two tails, χ^2^ test or Exact Fisher test, as appropriate. To evaluate the association of demographic and clinical characteristics at baseline and the occurrence of any adverse event after the second vaccine dose and after booster dose, univariable and multivariable logistic regression analyses were performed. In multivariable logistic regression, variables with *p* < 0.05 at univariable analysis were included. Statistical tests were considered significant at *p* < 0.05. Data were analyzed using Stata (StataCorp. 2021. Stata Statistical Software: Release 17. StataCorp LLC, College Station, TX, USA).

## 3. Results

### 3.1. Patients

In total, 78 RA patients were assessed at the time of the first vaccine dose and followed-up after the second vaccine dose; however, only 47 of them were followed-up after the third dose. Demographic, clinical, and life-style factors of the patients are specified in Table 1. The median DAS28CRP at T0 was 2.7 (IQR 2.1–3.4). Overall, 36% of patients were in clinical remission, 31% had low-disease activity, 33% moderate activity and no patient had a highly active disease. A total of 56% of the patients were receiving conventional synthetic disease-modifying anti-rheumatic drugs (csDMARDs; mainly methotrexate); 16.7% were receiving tumor necrosis factor (TNF)α-inhibitors, 48.7% other bDMARDs and 5.1% Janus kinase (JAK)-inhibitors. 

Thirty-two percent of patients were receiving a combination of cs and bDMARDs. Finally, 43.6% were treated with low-dose corticosteroids (<7.5 mg/day of prednisone or equivalent) in combination to cs and bDMARDs.

### 3.2. Adverse Events

AEs were observed in 40/78 (51.3%) and 13/47 (27.7%) patients at T1 and T2, respectively (*p* < 0.002). However, it has to be underlined that only 19/40 (47.5%) patients who had AEs after the second dose received the third dose, versus 28/38 (74%) of those who did not suffer any AEs at T1 (*p* < 0.004). Only seven patients (9%) had AEs at both doses. Pain at the injection site (28.2%) was the most frequent AE at T1, followed by systemic AEs: fatigue (23.1%), arthralgia (19.2%), fever (15.4%), headache (11.5%) and myalgia (10.3%). At T2, a reduction in total AEs, that was significant for pain at injection site (*p* < 0.007) and arthralgia (*p* < 0.004), was observed (Figure 1). No SAEs were reported at both T1 and T2. We then evaluated possible associations between baseline parameters and the occurrence of AEs. At T1, receiving glucocorticoids was significantly associated (*p* = 0.039) with the occurrence of AEs, whereas receiving other bDMARDs and being vaccinated for seasonal influenza were inversely associated (*p* = 0.044 and *p* = 0.013, respectively) with the occurrence of AEs at univariable analysis; however, only vaccination for seasonal influenza maintained a significant difference (*p* = 0.027) when adjusted in the multivariable analysis (Table 2). No significant associations were found at T2. 

### 3.3. Disease Flares

There were eight disease flares at T1 (10.3%) and 6 at T2 (12.8%). Flares occurred during the first week after vaccination, and were generally mild and not clinically relevant. Indeed, although DAS28CRP slightly increased at T1 and T2 (Figure 2), it remained under the cut-off of high disease activity in all patients, and only two patients required changes in ongoing immunosuppressive therapy. 

Ongoing glucocorticoid therapy was the only baseline parameter significantly associated with the occurrence of flares; in particular, we did not find associations between having a higher disease activity at baseline and the onset of flares (Table 3). Furthermore, patients who had flares did not suspend immunosuppressive therapy during vaccine administration.

According to the post hoc power analyses performed on the sample available to study outcomes after the second dose, only flu vaccination showed a quite good power, 73.5%, in detecting differences in proportions of AEs, and treatment with glucocorticoids a power of 77.7% in detecting difference in proportions of flares. 

### 3.4. Autoantibodies

Six out of thirty patients (20%) were positive for low-titer ANA (≤1:160) at baseline.

A slight, not significant, increase in ANA titer (from 1:80 to 1:160) was found in three patients after the second dose. However, the titer remained stable after the third dose and ANA were not associated with the onset of disease flares or AEs. No new ANA appearance was found. 

Twenty-four out of thirty patients (80%) were positive for RF at baseline and no modifications have been observed at T1 and T2.

### 3.5. Effectiveness

After the first cycle, only 1 in 78 (1.3%) patients presented a breakthrough infection (BI), versus 9 in 47 (19.2%) after the third dose, which all occurred during the first month of 2022 (*p* = 0.001). All infections were mild with the exception of one that was moderate; however, no patient required hospitalization (Table 4). No significant associations were found between baseline parameters and the occurrence of BI.

## 4. Discussion

The worldwide coverage for vaccine-preventable diseases in ARD patients is low, despite the increased risk of serious infections and related complications. Both the concern of impaired vaccine immunogenicity caused by immunosuppressive therapy and the fear of disease flares due to the intrinsic overreactive immune system have always limited this good clinical practice in ARD patients with the traditional vaccines [17]. 

At first, COVID-19 vaccines were seen even more cautiously, due to their new technology and lack of ARDs among populations evaluated in the pivotal trials [18]. However, in the last two years, vaccination was demonstrated to be a successful strategy to control COVID-19 pandemic both in the general population and in ARDs. Several real-life studies showed that some immunosuppressive agents decrease the titer of specific anti-RBD antibodies, but the large majority of patients can still develop a good immune response [8,10,43].

On the other hand, data on vaccine safety in ARDs are scarce and affected by several limitations. Indeed, most studies are conducted on heterogeneous populations of different ARDs, are focused on the first vaccine cycle, have a retrospective design, the occurrence of disease flares has been inferred through survey, and only one study evaluated the immunological safety (new autoantibodies appearance) [8,19,20,21,22,23,24,25,26,27,28,29,30,31,32,33,34,35,36,37,38,39,40,44].

Here, the safety of anti-Severe Acute Respiratory Syndrome-CoronaVirus (SARS-CoV)-2 BNT162b2 vaccine after the first cycle and the third dose in a cohort of RA patients was longitudinally analyzed. The vaccine showed an overall good safety profile with no patient reporting SAEs, and a percentage of total AEs similar to that reported in the general population in large trials [45] that does not increase after the third dose. 

The relatively low rate of AEs observed in our population also confirms data from previous studies in ARD patients after the first vaccine cycle. Tang et al. have recently systematically reviewed these data from 47 selected studies on a total of 4433 ARD patients after the first and second dose. The rate of SAEs was very low (0.04%). Local pain was the most common AE, followed by fatigue, arthro-myalgia and headache [8]. The large international registry EULAR COVAX showed 37% of AEs among a cohort of 4600 patients affected by ARDs after the first cycle, versus 40% observed in the control group of non-rheumatic inflammatory diseases. The prevalence of SAEs was low (0.4%), and the most frequent AEs were similar to those reported by the review of Tang et al. (pain at injection site, fatigue, arthro-myalgia and fever) [20]. However, in some studies [36,37], AEs were not significantly different from those observed in normal controls, thus suggesting a direct mRNA vaccine responsibility in the AEs induction, independently of ARD.

We observed 10.3% of flares after the end of the vaccination cycle and 12.8% after the third dose. These data are in agreement with previous studies which reported a general low incidence of disease flares in ARD patients with most being mild/moderate [40]. In particular, data were not dissimilar from the 7.8% found by Bixio et al. after the first vaccine cycle on 77 RA patients, studied with the same methodological approach used in the current study, all in clinical remission [28], and the 16% observed by Syversen et al., after the third dose in a cohort of rheumatologic and bowel autoimmune diseases [29]. The treatment with glucocorticoids was significantly associated with a higher rate of AEs (Table 2) and flares (Table 3) at univariable logistic regression, but the significance was no more maintained after a multivariable analysis. The direct association between steroids and flares, which has already been described by Pinte et al. [21] and Connolly et al. [35], has been interpreted as due to more severe disease, thus needing steroids. However, the responsibility of COVID-19 vaccine in the induction of flare-ups has been questioned, since no difference in flare rates has been found between vaccinated and non-vaccinated autoimmune patients by some authors [21,30]. Rather, the recommendation of interrupting the immunosuppressive therapy before vaccination, has been suggested as a possible cause of flares, but data are still contradictory [40]. Indeed, some studies did not find any significant increase in flares or disease activity after temporary interruption of immunosuppressive therapy [31,32], whereas Araujo et al. have recently reported higher risk of flares in RA patients who interrupted methotrexate for two weeks during COVID-19 vaccination [46]. Another historical question is whether a higher baseline disease activity can affect vaccine safety. Indeed, before the pandemic, it was generally recommended that vaccination in patients with ARDs should be preferably administered during quiescent disease [47]. In apparent contrast, due to the severity of the pandemic emergency, the ACR recommendations on COVID-19 stated that vaccination should be performed in every ARD patient irrespective of disease activity [3,7]. In the current study, we did not find any significant association between the incidence of AEs or disease flares and a higher disease activity at baseline. On the other hand, being vaccinated for influenza was inversely associated with the onset of AEs after the second dose. The safety of the influenza vaccine in RA patients has been known for a long-time [17], even when the adjuvanted flu vaccine was used [48]. However, it is difficult to provide an interpretation of this unexpected result, which may probably be explained only in the light of the non-specific effects of the influenza vaccine, particularly when combined with adjuvants [49], in the cross-protection against COVID-19 [50] or respiratory syncytial virus [51]. The non-specific cross-protection against COVID-19 may possibly be extended to better tolerance of COVID-19 vaccine. Alternatively, it may be hypothesized that people who are used to being vaccinated are naturally selected as those who may better tolerate vaccinations, thus being less prone to develop AEs. Whatever the involved mechanism, to the best of our knowledge, this is the first time that this type of potentially predictive factor of tolerance to COVID-19 vaccine in RA patients, has been identified. In fact, previously, only stable disease had been described as a negative predictive factor of post-vaccine flares in ARDs [34].

Very few studies addressed vaccine safety after the third dose. Assawasaksakul et al. have prospectively evaluated a small population of systemic lupus erythematosus (SLE) and RA patients. They found no SAEs, and the three most common AEs were injection site pain, fatigue and fever [52]. Even in the study by Syversen et al. mentioned above, the rate of AEs after the third vaccine dose was low and comparable to that observed after the second dose [29].

However, analyzing the rate of AEs in patients affected by autoimmune arthritis is a challenge for clinicians. Indeed, it should be taken into account that the large majority of systemic AEs to vaccines (i.e., arthralgia, myalgia, fatigue, fever) can resemble a disease flare and vice versa. As reported above, we tried to discriminate between AEs and flares, through the DAS28CRP, a universally accepted index which integrates both subjective (tender joints, global health score on a visual analogic scale) and objective (swollen joints and CRP) variables. However, it is impossible to exclude that some of the systemic AEs are not true AEs, rather mild disease flares, under the threshold to be detected by DAS28CRP modifications. 

The clinical safety was parallel to immunological safety, as inferred by slight, not significant, ANA titer modifications, without new autoantibody appearance, thus confirming the original result of Blank et al. in inflammatory arthritis after BNT162b2 vaccine [44] and our previous observations in RA patients and healthy subjects after common vaccines [48,53].

Both vaccine components, antigens and adjuvants, might induce autoimmunity onset and flares by different mechanisms, such as molecular mimicry, bystander activation and polyclonal activation [15,16]. BNT162b2 COVID-19 vaccine is an mRNA vaccine, whose composition includes in the same molecule not only the information for antigen synthesis, but even the adjuvant activity, which is exerted through the cellular stimulation of the pattern-recognition receptors, thus raising concern that in ARD patients with high disease activity, the adjuvant-induced activation of innate immunity and oxidative stress may induce autoantibody production and flares [40,54].

The effectiveness of BNT162b2 COVID-19 vaccine in preventing infection was excellent at T1, with breakthrough mild COVID-19 infection occurring in only 1/78 (1.3%). After the third dose, the rate of BI worsened, with 9/47 infected patients (19.2%; 8 mild and 1 moderate), probably due to the spreading of the Omicron variant of concern, which mostly escapes recognition by original vaccine [55] and became predominantly in Italy less than 1 month after its first detection, representing on 3 January 2022 76.9–80.2% of notified SARS-CoV-2 infections [56].

This hypothesis is corroborated by the overall good immunogenicity of the original SARS-CoV-2 strain, emerged by our previous studies conducted on a part of these patients. Indeed, we already showed that the antibody specific response was present in almost all subjects after the first cycle and was significantly strengthened after the third dose of BNT162b2 mRNA vaccine [10,43].

However, recently, a very promising treatment for COVID-19 has been set up, including the chimeric antigen receptor (CAR) T-cells, which may be useful and effective not only in oncology, but even in autoimmune and viral diseases [57].

The main limits of this study are the relatively low number of recruited patients and the lack of control groups, represented by non-vaccinated RA patients and vaccinated normal controls, that may affect the robustness of these results. Moreover, considering the multicentric real-life design of the study, reporting or investigator bias cannot be excluded, thus reducing the generalizability of results.

Conversely, some strengths of the study are its prospective design, the presence of a homogeneous disease cohort of RA patients, followed-up from the first cycle to the third dose, the evaluation of disease activity by clinical examination and the analysis of immunological safety, which allow the reliability of data be increased. 

## 5. Conclusions

In conclusion, the current preliminary study confirms a good safety profile of the BNT162b2 COVID-19 vaccine, at both the first cycle and the third dose, in RA patients under immunosuppressive therapy. No SAEs occurred, and the rate of AEs and disease flares was low.

For the first time, to the best of our knowledge, evidence was provided that being vaccinated for influenza was inversely associated with the onset of AEs after the second dose, thus allowing identification of a predictive factor of vaccine tolerance, should this result be confirmed in larger patient populations. A higher disease activity at baseline was not associated with a higher incidence of AEs or disease flares. The reduced vaccine effectiveness at T2 is a probable expression of the higher capacity of the Omicron variants to escape vaccine recognition. A future larger, controlled, study will allow to obtain definitive data on the mRNA COVID-19 vaccine safety in remitted and low-moderate disease activity RA patients.

## Figures and Tables

**Figure 1 biomedicines-11-00687-f001:**
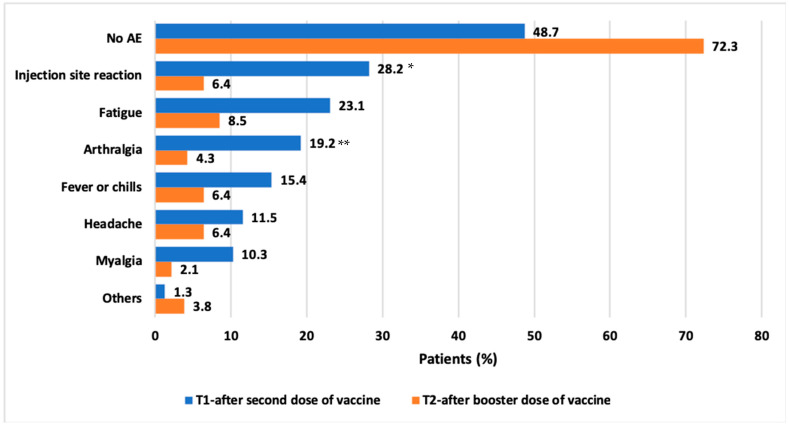
Rate of adverse events following the second dose (78 patients) and the booster dose (47 patients) COVID-19 vaccination in patients with rheumatoid arthritis. * *p* < 0.007; ** *p* < 0.004.

**Figure 2 biomedicines-11-00687-f002:**
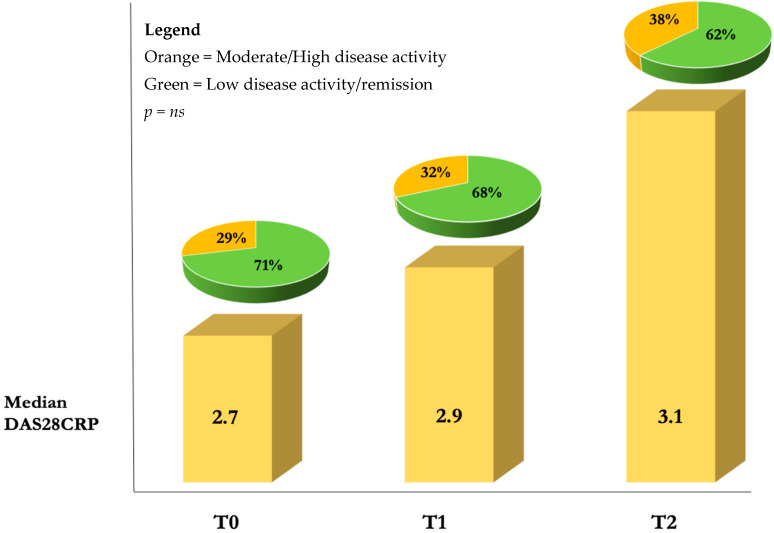
Disease activity score in the rheumatoid arthritis patients at baseline and during the follow-up.

**Table 1 biomedicines-11-00687-t001:** Demographic and clinical characteristics of 78 patients with rheumatoid arthritis enrolled in the study.

		n (%)
**Total**		78 (100)
**Age** median (IQR) years		61 (55–67)
**Gender** Female		60 (76.9)
**Comorbidities**		
None		31 (39.7)
One comorbidity		26 (33.3)
≥2 comorbidities		21 (26.9)
**BMI** (n = 66), median (IQR)		25 (21–28)
**Smoking habits**		
Yes		16 (20.5)
No		62 (79.5)
**Flu vaccination**		
Yes		37 (47.4)
No		25 (32.1)
Unknown		16 (20.5)
**Disease duration** (n = 68), median (IQR) years		8 (5–16)
*** Antinuclear antibodies (ANA)**	positive	6 (20)
	negative	24 (80)
*** Rheumatoid factor (RF)**	positive	24 (80)
	negative	6 (20)
**Immunosuppressive drugs, regimens containing**		
csDMARDs		44 (56.4)
TNF-inhibitors		13 (16.7)
Other bDMARDs		38 (48.7)
JAK-inhibitors		4 (5.1)
Glucocorticoids		34 (43.6)
**Disease activity, time-point T0**		
DAS28CRP, median (IQR)		2.7 (2.1–3.4)
Remission (DAS28 < 2.6)		29 (37.2)
Low (2.6 ≤ DAS28 ≤ 3.2)		26 (33.3)
Moderate (3.2 < DAS28 ≤ 5.1)		23 (29.5)
High (DAS28 > 5.1)		0

IQR = interquartile range; BMI = Body mass index; csDMARDs = conventional synthetic disease- modifying anti-rheumatic drugs; bDMARDs = biologic DMARDs; DASCRP = Disease Activity Score-C reactive protein; TNF = Tumor necrosis factor; JAK = Janus Kinase. * As reported in the text, ANA and RF evaluation was performed in a subgroup of 30 patients.

**Table 2 biomedicines-11-00687-t002:** Univariable and multivariable logistic regression of adverse events at T1 and T2 according to demographic and clinical variables.

Characteristics	Adverse Events Time–Point T1 *						Adverse Events Time–Point T2 *				
	Yes n (%)	No n (%)	Logistic Regression				Total n (%)	Yes n (%)	No n (%)	Logistic Regression	
			OR (95% CI)	*p*	aOR (95% CI)	*p*				OR (95% CI)	*p*
**Total**	40 (51.3)	38 (48.7)					47	13 (27.7)	34 (72.3)		
**Age** Median (IQR) years	60 (55–67)	62 (55–68)	0.94 (0.65–1.36)	0.755			61 (55–69)	61 (55–63)	63 (58–72)	0.76 (0.44–1.31)	0.326
**Gender** Male	6 (15.0)	12 (31.6)	1				13 (27.7)	3 (23.1)	10 (29.4)	1	
Female	34 (85.0)	26 (68.4)	2.61 (0.87–7.90)	0.088			34 (72.3)	10 (76.9)	24 (70.6)	1.39 (0.31–6.14)	0.665
**Comorbidities**											
None	17 (42.5)	14 (36.8)	1				21 (44.7)	7 (53.8)	14 (41.2)	1	
One comorbidity	9 (22.5)	17 (44.7)	0.43 (0.15–1.27)	0.130			16 (34.0)	5 (38.5)	11 (32.3)	0.91 (0.22–3.7)	0.893
≥2 comorbidities	14 (35.0)	7 (18.4)	1.64 (0.52–5.20)	0.395			10 (21.3)	1 (7.7)	9 (26.5)	0.22 (0.2–2.12)	0.191
**BMI** Median (IQR)	24 (21–28)	25 (23–28)	0.95 (0.86–1.07)	0.478			24.6 (22.5–28)	26.2(23.3–28.4)	24.2 (21.6–28)	1.16 (0.94–1.43)	0.153
**Smoking habits** No	32 (80.0)	30 (78.9)	1				33 (70.2)	9 (69.2)	24 (70.6)	1	
Yes	8 (20.0)	8 (21.1)	0.94 (0.31–2.81)	0.908			14 (29.8)	4 (30.8)	10 (29.4)	1.07 (0.27–4.28)	0.927
**Flu vaccination** Yes	13 (32.5)	24 (63.2)	1		1		24 (51.1)	7 (53.8)	17 (50.0)	1	
No	17 (42.5)	8 (21.2)	3.92 (1.33–11.52)	0.013	3.46 (1.14–10.4)	0.027	13 (27.7)	4 (30.8)	9 (26.5)	1.08 (0.25–4.70)	0.919
Unknown	10 (25.0)	6 (15.8)	3.07 (0.91–10.39)	0.070	3.11 (0.88–10.9)	0.077	10 (21.3)	2 (15.4)	8 (23.5)	0.61 (0.10–3.61)	0.583
**Disease duration** (med)	9 (4–15)	7 (5–17)	1.01 (0.96–1.06)	0.714			9 (5–17)	8 (5–16)	9 (5–17)	1.00 (0.93–1.07)	0.972
**Immunosuppressive drugs, regimens containing**											
csDMARDs	23 (57.5)	21 (55.7)	1.10 (0.45–2.68)	0.842			27 (57.5)	8 (61.5)	19 (55.9)	1.26 (0.34–4.66)	0.726
TNF-inhibitors	9 (22.5)	4 (10.3)	2.47 (0.69–8.82)	0.165			6 (12.8)	3 (23.1)	3 (8.8)	3.1 (0.54–17.9)	0.206
Other bDMARDs	15 (37.5)	23 (60.5)	0.39 (0.16–0.97)	0.044	1.25 (0.47–3.32)	0.648	31 (66.0)	6 (46.1)	25 (73.5)	0.31 (0.08–1.17)	0.083
JAK-inhibitors	3 (7.5)	1 (2.6)	3.00 (0.30–30.2)	0.351			0				
Glucocorticoids	22 (55.0)	12 (31.6)	2.35 (1.05–6.68)	0.039	2.37 (0.90–6.27)	0.082	16 (34.0)	6 (46.2)	10 (29.4)	2.06 (0.55–7.67)	0.283
**Disease activity DAS28CRP**											
Remission/Low	27 (67.5)	28 (73.7)	1				30 (63.8)	7 (53.8)	23 (67.6)	1	
Moderate/High	13 (32.5)	10 (26.3)	1.35 (0.51–3.59)	0.550			17 (36.2)	6 (46.2)	11 (32.4)	1.79 (0.49–6.61)	0.381

* Pain, redness or swelling at the site of injection; generalized muscle or joint pain, headache, fever, chills, fatigue or vomiting; IQR = Interquartile range; OR = Odds Ratio from univariable logistic regression; CI = Confidence interval; aOR = OR adjusted for variables with *p* < 0.05 at univariable analysis (flu vaccination, therapies containing other bDMARDs and glucocorticoids); BMI = Body mass index; csDMARDs = conventional synthetic disease- modifying anti-rheumatic drugs; bDMARDs = biologic DMARDs; TNF = Tumor necrosis factor; JAK = Janus Kinase; DAS28CRP = Disease activity score -C reactive protein.

**Table 3 biomedicines-11-00687-t003:** Univariable analysis of flares according to demographic and clinical characteristics.

Characteristics	Disease Flare					
	Time–Point T1			Time–Point T2		
	Yes n (%)	No n (%)	*p*	Yes n (%)	No n (%)	*p*
**Total**	8 (10.3)	70 (89.7)		6 (12.8)	41 (87.2)	
**Age** Median (IQR) years	64 (58–70)	60 (54–67)	0.369	60 (57–70)	61 (55–68)	0.962
**Gender** Male	0 (0)	18 (25.7)	0.187	2 (33.3)	11 (26.8)	1.000
Female	8 (100)	52 (74.3)		4 (66.7)	30 (73.2)	
**Comorbidities**						1.000
None	3 (37.5)	28 (40.0)	1.000	3 (50.0)	18 (43.9)	
One comorbidity	3 (37.5)	23 (32.9)		2 (33.3)	14 (34.2)	
≥2 comorbidities	2 (25.0)	19 (27.4)		1 (16.7)	9 (21.9)	
**BMI** Median (IQR)	25 (22–32)	25 (21–28)	0.371	24 (23–26)	25 (22–28)	0.883
**Smoking habits** No	8 (100.0)	54 (77.1)	0.195	4 (66.7)	29 (70.7)	1.000
Yes	0 (0)	16 (22.9)		2 (33.3)	12 (29.3)	
**Flu vaccination** No	3 (37.5)	28 (40.0)	0.795	3 (50.0)	10 (24.4)	0.373
Yes	3 (37.5)	23 (32.9)		3 (50.0)	21 (51.2)	
Unknown	2 (25.0)	19 (27.4)		0 (0)	10 (24.4)	
**Disease duration** Median (IQR) yr	8 (4–13)	8 (5–17)	0.475	5 (4–12)	10 (6–18)	0.110
**Immunosuppressive drugs, regimens containing:**						
csDMARDs	3 (37.5)	41 (58.6)	0.842	3 (50.0)	24 (58.5)	1.000
TNF-inhibitors	3 (37.5)	10 (14.3)	0.124	1 (16.7)	5 (12.2)	1.000
Other bDMARDs	4 (50.0)	34 (48.6)	1.000	2 (33.3)	29 (70.7)	1.161
JAK-inhibitors	0 (0)	4 (5.7)	1.000	0 (0)	0 (0)	
Glucocorticoids	7 (87.5)	27 (38.6)	0.018	2 (33.3)	14 (34.2)	1.000
**Disease activity T0**						
**DAS28CRP**, median (IQR)	2.7 (2.0–3.0)	2.8 (2.1–3.5)	0.442	2.4 (1.9–2.9)	3.1 (2.5–3.6)	0.061
Remission/Low	8 (100)	47 (67.1)	0.097	6 (100)	24 (58.5)	0.074
Moderate/High	0 (0)	23 (32.9)		0 (0)	17 (41.5)	

IQR = Interquartile range; BMI = Body mass index; csDMARDs = conventional synthetic disease- modifying anti-rheumatic drugs; bDMARDs = biologic DMARDs; TNF = Tumor necrosis factor; JAK=Janus Kinase; DAS28CRP = Disease activity score -C reactive protein.

**Table 4 biomedicines-11-00687-t004:** Breakthrough infection in the rheumatoid arthritis patients at T1 and T2.

		Time-Point T1 n (%)	Time-Point T2 n (%)
**Total**		78	47
**Infection**			
	Yes	1 (1.3)	9 (19.2) *
	No	77 (98.7)	38 (80.8)
**Severity**			
	Mild	1	8
	Moderate	0	1

* *p* < 0.002.

## Data Availability

Not applicable.

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
