# Peer review of "The Third Dose of BNT162b2 COVID-19 Vaccine Does Not “Boost” Disease Flares and Adverse Events in Patients with Rheumatoid Arthritis"

_biomedicines, 2023, doi:10.3390/biomedicines11030687_

Round 1

Reviewer 1 Report

Reviewer Comments

Article

The third dose of BNT162b2 COVID-19 vaccine does not 3 “boost” disease flares and adverse events in patients with rheumatoid arthritis

The authors cover an interesting hot topic which is booster dose in patients with rheumatoid arthritis.

Abstract

The abstract should include the statistical data including number ,percents and P value and so ,on ..

48.. COVID-19 complications, than the general. English editing and grammar revision are mandatory.

60… in ARDs patients. Please uniform this abbreviation

60…the research gap should be more specific to rheumatoid arthritis

Materials and methods …. The sample size should be calculated.

The section of materials and methods need to be improved …the authors have to determine what they evaluated at each stage or after each dose.

121.. Seventy-eight RA patients were evaluated after the second vaccine. This is in contrast to what have been mentioned in the methods section as the authors mentioned that they evaluated the patients with the first dose

131… (<7.5 mg/die…English editing is mandatory for the whole manuscript

135… (p=0.001631) …very strange number to represent P value ..it is uncommon to represent p value with 7 numbers

P value need to be adjusted at the whole manuscript to cope with the scientific writing

The writing of the results is to some extent unorganized, multiple total numbers which is to some extent confusing

Table 1... Smoking habits ... Unknown…this is a strange answer if all patients were evaluated.

Table 2.The authors should determine the side effects they asked about

Figure 1...others not other

P value is so strange

Limitations and strength of the study are mandatory.  

Author Response

  • The abstract should include the statistical data including number, percents and P value and so, on ..

These data have been added to the abstract

  • COVID-19 complications, than the general. English editing and grammar revision are mandatory.

English editing has been carried out.

  • 60… in ARDs patients. Please uniform this abbreviation

The abbreviation has been uniformed throughout the manuscript

  • 60…the research gap should be more specific to rheumatoid arthritis

It has been changed according to the suggestion

  • Materials and methods …. The sample size should be calculated.

We conducted post-hoc power calculations at a two-sided 5% significance level to evaluate the ability of the study to detect the difference observed between two groups.

  • The section of materials and methods need to be improved …the authors have to determine what they evaluated at each stage or after each dose

These points have been better clarified as requested

  • . Seventy-eight RA patients were evaluated after the second vaccine. This is in contrast to what have been mentioned in the methods section as the authors mentioned that they evaluated the patients with the first dose

It has been reported more clearly

  • 131… (<7.5 mg/die…English editing is mandatory for the whole manuscript

It has been corrected

  • 135… (p=0.001631) …very strange number to represent P value ..it is uncommon to represent p value with 7 numbers

It has been changed, by approximating in p<0.002.

  • P value need to be adjusted at the whole manuscript to cope with the scientific writing

The same approximation has been carried out throughout the manuscript for the other p.

  • The writing of the results is to some extent unorganized, multiple total numbers which is to some extent confusing

Results have been partially rewritten to be clearer

  • Table 1...Smoking habits ... Unknown…this is a strange answer if all patients were evaluated.

This information has been recovered and the tables have been modified consequently

  • Table 2. The authors should determine the side effects they asked about

The side effects asked about and reported in the text have also been reported in the Table 2, with a footnote.

  • Figure 1...others not other

It has been corrected

  • P value is so strange

It has been corrected

  • Limitations and strength of the study are mandatory.

Limitations and strengths of the manuscript have been expanded (lines 273-286).

Reviewer 2 Report

Thank you for the invitation to review this manuscript. The authors have made an interesting piece to underscore the side effects, its severity and impact of the vaccines on disease course. I have few suggestions;

The introduction section is very brief. I will suggest the authors incorporate information about the need for pharmacovigilance studies among diseased patients. The problem area should be highlighted here. Moreover, the authors should emphasize more on the reason why side effects monitoring is crucial for RA patients. how many studies have been conducted on this population before, what are the limitations of the previous studies and how the current study is covering up the limitations and literature gap.

I will suggest the authors to make the methodology section in sub-headings including ethics, study population and location, study design, sampling technique, sample size estimation, data collection form/data collection, components of data collection form and its reliability and validity, operational definitions used in the manuscript, statistical analysis.

The p-values up to three decimals are appropriate to be described in the manuscript. 

The authors can convert the figure to color format.

There is not only one limitation to this study. Adjustment of confounders, reporting or investigator bias, impact of disease on the severity of symptoms and impact of disease to link the symptoms with vaccines, generalizability of findings etc. are some other limitations. The authors are required to provide information on the strengths of this study. The low sample size is very major drawback of this study, how this sample will impact the results and how the authors direct future researchers about this limitation. Variations in the types of vaccines are other issues linked with homologous and heterologous vaccine impact on the side effects profile. 

How factors identified in the study linked with side effects profile, please provide plausible mechanisms for the factors. How influenza vaccination can reduce the severity or flareup following the shots of covid-19 vaccines.

These are a few questions that should be addressed in this manuscript. 

Author Response

We would like to thank the reviewer for his/her kind suggestions, which have allowed to markedly improve the manuscript.

In particular:

  • The introduction section is very brief. I will suggest the authors incorporate information about the need for pharmacovigilance studies among diseased patients. The problem area should be highlighted here. Moreover, the authors should emphasize more on the reason why side effects monitoring is crucial for RA patients. how many studies have been conducted on this population before, what are the limitations of the previous studies and how the current study is covering up the limitations and literature gap.

The Introduction has been partly modified, by adding a period (lines 75-101) for clearly explaining the issue of preventive anti-infectious vaccination in RA patients, the concern of inducing flares by vaccination and the number of patients already studied, with the limits in the published papers, which make the different patient populations poorly comparable.

  • I will suggest the authors to make the methodology section in sub-headings including ethics, study population and location, study design, sampling technique, sample size estimation, data collection form/data collection, components of data collection form and its reliability and validity, operational definitions used in the manuscript, statistical analysis.

This section has been revised as suggested, even though, due to the real-life observational design of the study, some points are missing such as reliability/validity tests (this limit has been added at the end of the discussion).

  • The p-values up to three decimals are appropriate to be described in the manuscript. 

We approximate the p to three numbers after the comma throughout the manuscript.

  • The authors can convert the figure to color format.

It has been modified

  • There is not only one limitation to this study. Adjustment of confounders, reporting or investigator bias, impact of disease on the severity of symptoms and impact of disease to link the symptoms with vaccines, generalizability of findings etc. are some other limitations. The authors are required to provide information on the strengths of this study. The low sample size is very major drawback of this study, how this sample will impact the results and how the authors direct future researchers about this limitation. Variations in the types of vaccines are other issues linked with homologous and heterologous vaccine impact on the side effects profile. 

The section of limitations and strengths of the manuscript has been changed according to the right and constructive suggestions of the reviewer (lines 273-286).

  • How factors identified in the study linked with side effects profile, please provide plausible mechanisms for the factors. How influenza vaccination can reduce the severity or flareup following the shots of covid-19 vaccines.

The reviewer is right even in this remark. However, we already tried to interpret how the influenza vaccination can reduce the severity or flare up following the shots of COVID-19 vaccines (lines 174-186) and we did not change this part, because we have no alternative hypotheses for explaining this inverse association.

Reviewer 3 Report

Andrea Picchianti Diamanti and colleagues present a quality and well-written experimental manuscript focused on the third dose of BNT162b2 COVID-19 vaccine does not “boost” disease flares and adverse events in patients with rheumatoid arthritis.

Authors aimed to analyze the clinical and immunological safety of BNT162b2 vaccine, in a cohort of rheumatoid arthritis patients followed-up from the first vaccine cycle to the third dose. 

Authors suggest that vaccine showed an overall good safety profile with no patient reporting serious adverse events, and a low percentage of total adverse events at both doses. Flares were observed in 10.3% of patients after the end of the vaccination cycle and 12.8% after the third dose. Being vaccinated for influenza was inversely associated with the onset of adverse events after the second dose, at both univariable and multivariable analysis. This result could allow to identify a predictive factor of vaccine tolerance, if confirmed in larger patient populations. Neither a higher disease activity at baseline nor having suspended immunosuppressive agents during vaccine administration, were associated with a higher incidence of adverse events or disease flares. Effectiveness was excellent after the second dose, with only 1 mild breakthrough infection and got worse after the third dose, as a probable ex- pression of the higher capacity of the Omicron variants to escape vaccine recognition.

Authors found that being vaccinated for influenza was inversely associated with the onset of adverse events after the second dose, thus allowing to identify a predictive factor of vaccine tolerance, should this result be confirmed in larger patient populations. 

Finally, authors conclude that the current study confirms a good safety profile of BNT162b2 COVID-19 vaccine, at both the first cycle and the third dose, in rheumatoid arthritis patients under immunosuppressive therapy. No serious adverse events occurred, and the rate of adverse events and disease flares was low.

Overall, the manuscript is valuable for the scientific community and should be accepted for publication after edits are made.

===========================

Other comments:

1) Please check for typos throughout the manuscript.

2) With regards to novel treatments of COVID-19 – authors are kindly encouraged to cite the following review that describes cell based approached for such therapy. DOI: 10.3390/biomedicines9010059

Author Response

We would like to thank the reviewer for his/her kind suggestions, which have allowed to markedly improve the manuscript.

In particular:

  • Please check for typos throughout the manuscript

We have checked for typos throughout the manuscript.

  • With regards to novel treatments of COVID-19 – authors are kindly encouraged to cite the following review that describes cell based approached for such therapy. DOI: 10.3390/biomedicines9010059

The suggested reference has been included as reference 57 and the added phrase for including the manuscript is in the Discussion, lines 270-272.

Round 2

Reviewer 1 Report

The manuscript has been improved greatly 

Thanks for the authors